# Revisiting the Effect of Pyrolysis Temperature and Type of Activation on the Performance of Carbon Electrodes in an Electrochemical Capacitor

**DOI:** 10.3390/ma15072431

**Published:** 2022-03-25

**Authors:** Madhav P. Chavhan, Vaclav Slovak, Gabriela Zelenkova, Damir Dominko

**Affiliations:** 1Faculty of Science, University of Ostrava, 30. dubna 22, 701 03 Ostrava, Czech Republic; vaclav.slovak@osu.cz (V.S.); gabriela.zelenkova@osu.cz (G.Z.); 2Institute of Physics, Bijenička Cesta 46, HR-10000 Zagreb, Croatia; ddominko@ifs.hr

**Keywords:** supercapacitor, carbon xerogel, carbonisation, activation, surface oxygen functionality, electrical conductivity, electrochemical performance, meso-micropore connectivity

## Abstract

Hierarchical porous carbons are known to enhance the electrochemical features of electrodes in electrochemical capacitors. However, the contribution of surface oxygen and the resulting functionalities and wettability, along with the role of electrical conductivity and degree of amorphous or crystalline nature in the micro-mesoporous carbons, are not yet clear. This article considers the effect of carbonisation temperature (500–900 °C) and the type of activation (CO_2_, KOH) on the properties mentioned above in case of carbon xerogels (CXs) to understand the resulting electrochemical performances. Depending on the carbonisation temperature, CX materials differ in micropore surface area (722–1078 m^2^ g^−1^) while retaining a mesopore surface area ~300 m^2^ g^−1^, oxygen content (3–15%, surface oxygen 0–7%), surface functionalities, electrical conductivity (7 × 10^−6^–8 S m^−1^), and degree of amorphous or crystalline nature. Based on the results, electrochemical performances depend primarily on electrical conductivity, followed by surface oxygen content and meso-micropore connectivity. The way of activation using a varied extent of CO_2_ exposure and KOH concentrations played differently in CX in terms of pore connectivity from meso- to micropores and their contributions and degree of oxidation, and resulted in different electrochemical behaviours. Such performances of activated CXs depend solely on micro-mesopore features.

## 1. Introduction

The functionality and effectiveness of electrochemical capacitors (ECs) is closely related to the used electrode materials. To date, a wide range of active materials have been tested in EC applications, mostly based on transition metal oxides and sulphides (e.g., Co_3_O_4_, MnO_2_, WO_3_, NiO, CoNi_2_S_4_, Ni_3_S_2_) or porous carbons (biomass-based activated carbons, carbon gels with controllable porosity, graphene based materials, etc.) [1]. Carbonaceous materials, in general, offer a unique combination of properties, like high chemical and thermal stability, tuneable surface properties (surface area, pore size distribution, surface chemistry), electric conductivity, and low cost. In addition, they can be fabricated in various physical forms (powders, thin layers, wires, monoliths), which simplifies their assembly in ECs construction.

In recent years, research on the development of carbon-based electrodes for ECs has focused on improving their surface area with a proper pore hierarchy and incorporating heteroatoms or functionalities into the carbon structure [2,3,4,5,6]. As most of the surface area for double-layer charge storage is held inside micropores (<2 nm pore size), the pore hierarchy or connectivity between meso-micropores plays a vital role in the rapid transport of electrolyte ions to the internal surface [7]. The introduction of surface functionality using heteroatoms (e.g., N, S, P, O, B) is another way to enhance charge storage in carbon electrodes [8,9]. Such functionalities contribute to increase surface polarity, wettability, electric conductivity, and electron donor/acceptor properties, and also act as active sites for pseudocapacitance [2]. The incorporation of these functionalities into carbon through the precursor itself offers feasibility with a uniform distribution of functionalities over the carbon network and an improvement in bulk properties compared to other post-treatments used for functionalisation [6].

Carbons obtained from the sol-gel polymerisation of monomers, followed by solvent removal (drying step) and pyrolysis (also called carbonisation), are extremely popular as electrode materials for ECs, because of the tailorable pore size distributions, high surface area, and favourable surface chemistry [3,4,5,6,10]. Such properties are easily controlled through the cross-linked chemistry and the choice of operating parameters during processing steps. Sol-gel chemistry involves the addition and condensation reaction between hydroxylated benzenes (e.g., phenol, cresol, resorcinol, hydroquinone) and aldehydes (e.g., formaldehyde, furfural) to form a three-dimensional cross-linked polymeric gel network [6,10]. Particularly, carbon derived from the resorcinol-formaldehyde (RF) polymeric gel is widely studied as an electrode material due to its large surface area that stores charges through double layer formation, and the favourable pore hierarchy provides a rapid transport of ions. The tailoring of micro-mesoporous structure in RF-based carbon electrodes for EC applications is mostly investigated through synthesis steps by optimising concentrations of monomer, catalyst, and diluents, etc. [11,12], controlling the sol-gel transition period [11], or maintaining the pH of the initial precursor sol [11,12], drying protocols [7,13,14], solvent exchange steps [12], and activation protocols [7,15,16]. However, there is not much investigation in the prior art on the carbonisation stage that shows how the porous texture with pore connectivity, surface functionality, electrical conductivity, or degree of amorphous/crystalline nature arises at different carbonisation temperatures and their resulting performances when used as electrodes for ECs.

Previous studies on the pyrolysis of organic RF gel have shown a significant influence on carbon pore structure after pyrolysis temperatures [17,18,19,20]. More micropore development occurs after carbonisation, which is responsible for the high surface area in carbons. For carbonisation, low heating rates (<10 °C min^−1^) or several stages of heating with very low heating rates (1.7–5 °C min^−1^) were used to preserve the porosity of the carbonaceous product [19,21]. Furthermore, the particle size of the organic gel has found an intense effect on the surface area, which increases for the carbonisation temperature from 750 to 900 °C for an average particle size (dp) <212 µm [19]. In addition, a carbonaceous product emerges with different oxygen functionalities that significantly alter acidic-basic characteristics and surface interactions, which are responsible for the adsorption or improved wettability of electrolyte ions [2,3,6,22]. On the other hand, too many bulky oxygen groups can damage the sp^2^ carbon network and decrease electrical conductivity, ultimately hindering the charge transport in carbon electrodes [23,24]. Thus, an optimum oxygen content is desired for achieving excellent electrochemical performance. Even in the literature, the pyrolysis temperature was randomly chosen from 600–900 °C to make carbon electrodes for use in EC [10,25]. Therefore, a systematic investigation is necessary to understand the influence of the characteristics mentioned above on the final performance of carbon electrodes.

The charge storage in the carbonaceous network can be further enhanced through the creation of additional pores and the addition of oxygen functionality. Such pore structure in carbon can be easily created through intercalation or surface etching by physical or chemical activation [7,21,26]. In physical activation, the sample is exposed to gas phase oxidants, e.g., oxygen, air, steam, CO_2_, ozone, and nitrogen oxide at elevated temperature, while the chemical activation involves contacting the sample with the liquid phase as oxidants, e.g., nitric acid, hydrogen peroxide, ammonium persulfate, phosphoric acid, KOH and NaOH as alkali solutions [10]. During activation, oxygenated surface groups (e.g., carboxylic, anhydrides, lactones, quinones, and phenols) are introduced into the carbon surface. Thus, the relative contributions of surface oxygen functionality (or amount of total oxygen) and porous structure arising from meso-micropores in CXs will vary with the type of activation.

Based on the knowledge gap described above, the objectives of the present study are (i) to demonstrate the effect of the pyrolysis temperature from 500 °C to 900 °C on the properties of carbon xerogels (CXs) and their corresponding electrochemical behaviour for use in EC, and (ii) to explore the type of activation on CX, which is suitable for enhanced charge storage. For the activation of pores in CX, varied extents of CO_2_ exposure and different concentrations of KOH pellets are considered in this study.

## 2. Materials and Methods

### 2.1. Synthesis of CX and Its Activated Forms

The sol-gel synthesis of RF-derived CX is reported earlier in numerous research works. In short, the homogeneous precursor solution of resorcinol (R, recrystallised) and formaldehyde (F, 37–41% *w*/*v* GR; stabilised by 8–14% methanol) with sodium carbonate (Na_2_CO_3_, anhydrous) as catalyst (C) and deionised water as diluent (D) was poured into glass vials, sealed and held for polymerisation at 70 °C in an oven. The composition of the RF precursor sol was as follows: molar ratios of R/F = 0.5, R/C = 300, and R/D = 0.018. The pH of the initial precursor sol was measured as 5.88. The solution was held for 24 h at a temperature of 70 °C to complete the polymerisation and curing of the resulting gel. The solvent from the wet gel was removed through ambient drying by keeping the gel sample in an oven (Fisher Scientific, Pardubice, Czech Republic) at 60 °C for 15–16 h, without any sample pre-treatment or any other solvent exchange step. Finally, CXs were obtained by the carbonisation of dried gel in a tubular furnace at different temperatures, from 500 °C to 900 °C, with a heating rate of 5 °C min^−1^. The samples were held at 1 h when the final carbonisation temperature was reached. The inert environment inside the tubular furnace was maintained using nitrogen gas with a flow rate of 0.1 L min^−1^. Two methods of activation on CX sample were employed through uses of CO_2_ gas and the KOH solution, respectively. A physical activation using CO_2_ was performed immediately after the end of the carbonisation process while switching the inflow from N_2_ to CO_2_ at the same flow rate over varied durations of 4 and 6 h. In the chemical activation with KOH, pulverised CXs were activated by mixing predetermined ratios of KOH pellets with CX samples as 4:1 and 6:1 in deionised water as the solvent. The mixtures were dried by stirring at 85 °C to remove water and then heated in a tubular furnace (HPservis, Rajhrad, Czech Republic) to a temperature of 750 °C at a rate of 5 °C min^−1^, with a holding time of 1 h, at this temperature under the inert environment of nitrogen, with a flow rate of 0.1 L min^−1^. The resulting samples were first immersed in a 37% HCl solution, heated at 85 °C for 30 min, and then filtered to remove the acid solution with Whatman^TM^ Grade 1 filter paper. The final products were washed several times to reach a pH around 6 and finally dried at 110 °C for 5 h in an oven. The activated samples are named as ‘Type of activation-CX-CO_2_ exposure time or KOH pellet concentration to CX’, for example CO_2_-CX-4h or KOH-CX-6:1.

### 2.2. Material Characterisation

The porous textures of CX samples were obtained from the nitrogen adsorption–desorption experiment at −196 °C using Quantachrome^®^ ASiQwin™ version 3.0 instruments (Quantachrome instruments, Boynton Beach, FL, USA). All samples were degassed at a temperature of 300 °C in a vacuum prior to measurements. The estimation of micropore (surface area (S_mic_) and volume (V_mic_)) and mesopore (surface area (S_meso_) and volume (V_meso_)) features, and the corresponding pore size distribution of CX samples, were developed using nonlocal density functional theory (DFT). This theory considers the slit-pore model for pore widths <2 nm, and the cylindrical model for pore widths >2 nm, and from which the combined pore volume (V_DFT_) and surface area (S_DFT_) were calculated. The morphology of CXs was observed using a FEI Nova NanoSEM 450 high-resolution scanning electron microscope (HRSEM) (Thermo Fisher Scientific, Waltham, MA, USA) equipped with an Everhart-Thornley secondary electron detector and a through-lens detector. For such measurements, the water suspensions of the CX samples were dropped on silicon chip substrates, each with dimensions of 5 mm × 5 mm, and allowed to dry at ambient temperature. SEM images of activated CX samples were captured using JSM-7610F FE-SEM equipment (JEOL, Tokyo, Japan). The elemental composition was obtained using the CHNS varioMACROcube analyser (Elementar, Langenselbold, Germany). The total oxygen percentage was then determined by subtracting the total element percentages. The Nicolet 6700 Fourier transform infrared (FTIR) spectrometer was used to detect functional groups present in the CXs. For this purpose, the FTIR spectra were collected in an attenuated total reflection mode on a single reflection diamond ATR crystal. The spectra were collected in the range of 400–4000 cm^−1^ from 128 scans with the resolution of 4 cm^−1^. Furthermore, three measurements were performed for each sample, and spectra were then averaged. The baseline correction was carried out for all spectra. The surface oxygen content in the CXs was determined by a TG-MS experiment using SetsysEvolution equipment (Setaram, Lyon, France), coupled with a QMG 700 quadrupole mass spectrometer (Pfeiffer, Asslar, Germany). The TG-MS curves were obtained from 15 °C to 1000 °C at a heating rate of 10 °C min^−1^ under the inert environment of Ar with a flow rate of 20 mL min^−1^. The wettability of CX samples was analysed by water vapour adsorption at low relative pressure (P/P_0_ ≈ 0.3) in a desiccator filled with a saturated solution of CaCl_2_. The relative humidity content in the desiccator was measured using a hygrometer, 608-H2, Testo, and was noted as 27.7%. X-ray diffraction (XRD) patterns were recorded using Bruker D2 (Bruker, Billerica, MA, U.S.A.) equipped with a conventional X-ray tube (CuKα radiation, 30 kV, 10 mA) with the LYNXEYE 1-dimensional detector. The primary divergence slit module width 0.6 mm, Soler module 2.5, Airscatter screen module 2 mm, Ni Kbeta filter 0.5 mm were used in the range 10–90°, step 0.00805°, and time per step 1 s. From the XRD data, the inter-layer spacing ($d_{002}$)*,* crystallite height ($L_{c}$), crystallite diameter ($L_{a}$), and average number of aromatic layers per carbon crystallite ($N_{av}$) were calculated using Equations (1)–(4), respectively [27,28].
(1)$$d_{002}=\frac{\lambda }{2{\;\sin\theta }_{002}}$$
(2)$$L_{c}=\frac{K_{c}\lambda }{\beta_{002}{\;\cos\theta }_{002}}$$
(3)$$L_{a}=\frac{K_{a}\lambda }{\beta_{100}{\;\cos\theta }_{100}}$$
(4)$$N_{av}=\frac{L_{c}}{d_{002}}+1$$
where λ is the wavelength of incident X-ray (1.5405 Å), $\theta_{002}$ and $\theta_{100}$ are the peak positions of (0 0 2) and (1 0 0) planes in degrees, respectively, $\beta_{002}$ and $\beta_{100}$ are the Full Width at Half Maximum (FWHM) of (0 0 2) and (1 0 0) diffraction peaks, respectively, and $K_{c}=0.89$ and $K_{a}=1.84$ [28,29]. For the electrical conductivity measurements, CX monoliths were made in disc-shaped form with a diameter ranging from 1.2 to 1.9 cm, and the thickness varied from 1.5 to 2 mm. We used two contact methods, with contacts made by silver paint at the disc bases. Keithley 6517A electrometer was used both as the voltage source and amperemeter. Voltages used did not exceed 20 V.

### 2.3. Electrochemical Measurements

CX electrodes were prepared by laying the acetone-based slurry of CX powders (90% by weight) with poly(methyl methacrylate) binder (10% by weight) over the Toray carbon papers (TGP-H-060). The Toray carbon paper has the following specifications: thickness 0.19 mm, bulk density 0.44 g cm^−3^, porosity 78%, thermal conductivity 1.7 W (m·K)^−1^, electrical resistivity 80 mΩ cm. The slurry was coated on one side of the carbon paper with an area of 1 × 1  cm^2^, and the coated electrodes were dried at 60 °C for 12 h in an oven. The mass loading of the active CX material was found to be around 4–5 mg. For the electrochemical measurements, a symmetric electrochemical cell was assembled using CX electrodes, separated by Whatman^TM^ filter paper (Grade 1). An aqueous solution of KOH (2 M) was used as an electrolyte. Cyclic voltammetry (CV), chronopotentiometry (CP), and electrochemical impedance spectroscopy (EIS) techniques were employed for electrochemical measurements using VersaSTAT 3 Potentiostat Galvanostat. The specific capacitance of a single CX electrode in a symmetric capacitor cell was calculated from CV and CP techniques using Equations (5) and (6), respectively [7,15].
(5)$$C_{s}=\frac{\int_{0}^{v}\mathrm{IdV}}{m\;\Delta V\;S}$$
(6)$$C_{s}=\frac{2{\;I\;t}_{d}}{m\;(\Delta V-\mathrm{IR})}$$
where C_s_ is the specific capacitance of a single electrode (F g^−1^), I is the current (A), ΔV is the applied potential window (V), m is the mass of active material of the single CX electrode (g), S is the scan rate (V s^−1^), t_d_ is the discharging time (s), and IR is the ohmic drop (V). The Coulombic efficiency (η), specific energy (E in Wh kg^−1^), and specific power (P in W kg^−1^) of the electrochemical cell were estimated using Equations (7)–(9), respectively [2,6].
(7)$$\eta =\frac{t_{d}}{t_{c}}\times 100\%$$
(8)$$E=\frac{1}{2}\frac{\;C\;{\left(V-\mathrm{IR}\right)}^{2}}{3.6}$$
(9)$$P=3600\;\frac{E}{t_{d}}$$
where t_c_ is the charging time (s), and C is the capacitance of the symmetric cell (F g^−1^).

## 3. Results and Discussion

### 3.1. Effect of Carbonisation Temperature

During carbonisation, organic gel is heated at the chosen temperature in an inert environment (e.g., N_2_, Ar, or He), wherein organic material decomposes by leaving volatile matters and retaining the carbonaceous skeleton of polymeric gel. Such a process is complex and may involve several reactions simultaneously, e.g., dehydrogenation, isomerisation, and condensation, which are responsible for chemical changes. The diffusion of volatile components or evolution of pyrolytic gases throughout the gel matrix are responsible for the final porous structure in the resulting carbon product. Such a porous structure and the pore connectivity from meso-micropores can be influenced by different pyrolysis temperatures and, in turn, decide the electrochemical performance of carbon electrodes. At the same time, different temperatures used during carbonisation affect the amount of heteroatoms and the resulting functionalities in the carbon network. In particular, the oxygen content and the nature of functionality have a different effect on wettability, conductivity, and the resulting redox activity. For example, the oxygen functional groups (C-O, and C=O) present on the outer surface or at the edge of the basal plane are responsible for acidic characteristics [6,30], which improved the hydrophilicity and rapid charge transport in the electrode. On the other hand, the basicity of basic oxygen functionalities or resonating electrons of the aromatic ring in carbon leads to a redox reaction from the faradic interaction with electrolyte ions [2,3]. Therefore, the relative contributions of the porous structure arising from the micro-mesopores, the amount of total, and the surface oxygen with the type of functionalities, electrical conductivity, wettability, and the resulting redox activity will vary with the carbonisation temperature and, in turn, influence the final performance of the carbon electrodes.

In this article, the composition of the RF solution, as stated in the previous section, was taken from a previous study based on a high surface area with a significant proportion of mesopores [11,12]. For this purpose, the cross-link density in the polymeric gel was controlled through a variation of the R/C and R/D molar ratios. Upon pyrolysis of such a polymeric gel at different temperatures, the internal volume shrinks to form an interconnected network of carbon nanoparticulates with significant fractions of micropores and mesopores. Figure 1a,b represents the CX network obtained at two different carbonisation temperatures, of 500 °C and 800 °C, respectively. Such interconnected networks with void spaces are ideally suited for an easier access of the electrolyte ions and for a rapid charge transport. The insight into the porous texture of carbonised samples was understood through the nitrogen adsorption–desorption experiments (Figure 1c,d and Table 1). All samples exhibit Type IVa isotherms with hysteresis representing the presence of mesopores (Figure 1c) [31]. Capillary condensation starts at a relative pressure (P/P_0_) ~0.6, and the similar nature of hysteresis that occurred in all of these samples, carbonised at different temperatures, indicates a similar width of the mesopores. This is further confirmed by the pore size distribution (Figure 1d), obtained using DFT. The mesopores are in the range from 3.5 to 14 nm for all samples, and these are retained with increasing carbonisation temperature. On the other hand, these samples exhibit a similar microporous structure with an increase in the total micropore volume with an increase in the carbonisation temperature. The larger uptake at low relative pressure indicates the development of micropores due to the removal of volatiles during the carbonisation process. The analysis of micro-mesopore characteristics could be understood very well through the values obtained from DFT (Table 1). Thus, the increase in micropore features and retention of mesopore features with carbonisation temperature are responsible for the enhanced surface area of CXs.

Table 1 also shows the elemental compositions of carbonised samples at different temperatures, as obtained from the CHNS analyses. The decrease in the percentages of oxygen and hydrogen appeared due to the removal of labile groups during the pyrolysis process. The elemental compositions of these samples follow the trend with an increase in the carbonisation temperature. The chemical structure and its changes in relation to the increasing carbonisation temperature were further analysed by FTIR spectroscopy (Figure 2). It is obvious that the carbon structure formed at 500 °C contains the highest amount of oxygen (also the oxygen surface groups) that will decrease with the increasing pyrolysis temperature. This trend is evident from the measured spectra (Figure 2). The studied carbonaceous samples exhibit peaks with absorption at 1620 cm^−1^, 1430 cm^−1^, and in the range of 1250–1000 cm^−1^, which are probably connected to the C=O stretching vibration in carbonyls, the C-H stretching vibration of asymmetric and symmetric -CH_2_ groups, and the -CO stretching in lactons, ethers, and phenols, respectively [32,33,34]. In addition, bands approximately at 1250 cm^−1^ and 1100 cm^−1^ could be associated with C-O-C stretching vibrations of methylene ether, an intermediate product of resorcinol and formaldehyde, which is formed during their condensation [34]. Furthermore, the characteristic vibrations of the out-of-plane groups of –CH are noticeable in the region from 900 cm^−1^ to 700 cm^−1^ [12].

The amount of surface oxygen and the proportions of functionality arising in CX were analysed through TG-MS experiments (Figure 3). For this purpose, the MS signals from the evolution of H_2_ (*m*/*z* = 2)_,_ H_2_O (*m*/*z* = 18)_,_ CO (*m*/*z* = 28), CH_4_ (*m*/*z* = 16), and CO_2_ (*m*/*z* = 44) were analysed for the determination of the amount of surface elements (C, H, O) [35]. The TG profiles (Figure 3a) show a decrease in the mass of the CX samples with temperature and follow a trend with an increase in the carbonisation temperature. The amount of mass loss is proportional to the amount of oxygen and hydrogen elements in CX. The surface oxygen present in CX, calculated from TG-MS data, follows a trend similar to the oxygen composition of CHNS analyses (Figure 4). However, no signals are detected in MS for CX, obtained at 900 °C due to the very low percentage of surface oxygen (Figure 3b–e). The amount of evolution of CO and H_2_O is higher compared to the amount of evolution of CO_2_, clearly indicating that the phenol (-OH) (in CX-500 and CX-600) and carbonyl (C=O) groups (in all samples) predominate. Such oxygen surface groups are mainly responsible for introducing faradic charge storage in carbon electrodes according to the following reactions [3,6].
(10)$$>C-\mathrm{OH}⟺>C=O+H^{+}+e^{-}$$
(11)$$>C=O+e^{-}⟺>C-O^{-}$$

The hydrophilicity arising from surface oxygen functionalities was further understood by the water-vapour adsorption experiment at a relative pressure ~0.3 (Figure 4). The obtained results, summarised in Figure 4, are comparable with the information obtained from the TG-MS analysis (and also the elemental analysis). Based on the TG-MS analysis (Figure 3), it can be noted that the carbon CX-500 contained a relatively large amount of phenolic surface groups together with some lactones (Figure 3d—the phenols are decomposed around 600 °C) in contrast with the other carbons (CX-600 probably contains also a small amount of phenols). The decomposition of phenolic groups of the CX-500 sample is evidenced by the large amount of water (see Figure 3c) evolved in the same temperature region. The presence of a significant content of phenolic groups in the CX-500 sample could explain the surprisingly lower uptake of water vapours (Figure 4). The large content of hydroxyl groups could react together on the inner surface, mutually saturating possible hydrogen bonding, and create a “barrier” preventing the water vapour from reacting with other (carbonyl) surface groups. After the decomposition/removal of hydroxyls (samples pyrolysed at 600 °C and more), the adsorption of water vapour simply follows the oxygen content trend (Figure 4).

The XRD spectra of carbonised CX samples at different temperatures are shown in Figure 5. Two broad peaks were observed, ascribed to the (0 0 2) and (1 0 0) planes, for the CX samples. These peaks seem to become more intense with an increase in carbonisation temperature. The former (0 0 2) peak starts to change from a flattened shape to a more intense one with carbonisation temperature, suggesting the ordering of the aromatic structure during carbonisation. In the literature, such a peak is referred to as relating the stacking of graphene sheets via van der Waals forces [27,36]. On the other hand, the planes (1 0 0) represent the degree of aromatic ring condensation that appeared with the increase in carbonisation temperature. In general, more intense peaks that arise with an increase in pyrolysis temperature indicate a higher crystalline character in CXs. A summary of the structural properties of CXs with carbonisation temperature is presented in Table 2. Practically, the values of crystallite height ($L_{c}$) and average number of aromatic layers per carbon crystallite ($N_{av}$) are similar, and thus there is no influence on the stacking of graphene layers with carbonisation temperature. However, the decrease in values of interlayer spacing ($d_{002}$) with carbonisation temperature indicate an increase in the number of stacked graphitic layers organised in the carbonaceous structure [27,29]. Moreover, the increase in crystallite diameter ($L_{a}$) confirms the improved crystallinity of CXs at a high carbonisation temperature. Another parameter, R*, used in the literature to distinguish the concentration of single graphene layers, shows an increase in values with carbonisation temperature, indicating the highest concentration of single layers in CXs [12,29].

The structural changes during the carbonisation process led to changes in electrical conductivity (Figure 6). The CX sample carbonised at 500 °C exhibits a very low conductivity, which increases four orders of magnitude by carbonising at 600 °C and again two orders of magnitude by carbonising at 700 °C and reaching a plateau thereafter. Note that one of the possible reasons for the plateau is the two contact method of measuring DC conductivity, which is the only possible one for the disc-shaped samples, and therefore limited from above by lead and contact resistance.

The electrochemical performances of CX electrodes in a symmetric cell with an aqueous KOH solution as an electrolyte are presented in Figure 7. Significant differences in electrical conductivity lead to different CV plot shapes (Figure 7a,b), from a leap shape to an ideal rectangular shape for CX, carbonised at a lower temperature (500 °C) to a high temperature (800 °C), respectively. The higher amount of oxygen content and the resulting functionality [6,15] at a low carbonisation temperature are responsible for such a distorted shape of the CV plot. The contribution of oxygen functionality in CX electrodes, carbonised at a high temperature can be seen from the CV plot obtained at a low scan rate (Figure 7c). The deviation in the anodic branch of the CV plot is mainly due to the pseudocapacitance provided by the phenolic and carbonyl groups, as analysed from TG-MS data. Such a quasi-rectangular shape observed from CV data confirms the excellent accessibility of electrolyte ions onto the electrode surface arising from meso-micropore connectivity and the hydrophilicity provided by surface oxygen functionalities. The comparison of CX electrodes (Figure 7c) shows a large area under the CV plot for the CX electrode carbonised at 800 °C, followed by the CX electrode carbonised at 700 °C. The increase in conductivity and improved micropore features with optimum surface oxygen functionality in these electrodes are the reasons behind their superior performance compared to other types of CX electrodes carbonised at low temperatures. However, the absence or very low content of surface oxygen in the CX electrode carbonised at 900 °C shows a decrease in performance, although it has extreme micro-mesopore features compared to the CX electrode carbonised at 700 °C and 800 °C. The rate capability plot (Figure 7d) shows a stable performance with scan rate, indicating a good reversibility of the CX electrodes. The distinguishing performance between CX electrodes carbonised at 700 °C and 800 °C is observed from these rate capability plots. Similar observations in electrochemical performance are observed in the CP plots obtained between the same voltage range (Figure 7e). The deviation in the regular triangular shape is manifested through the oxygen functionality, which provides additional pseudocapacitance in addition to improved wettability [2,6]. A large IR drop is more significant in CX electrodes, carbonised at 500 °C and 600 °C due to more proportions of surface oxygen/functionality, as well as the low conductivity, which hinders the charge transport process. The longest charge–discharge time is observed for the CX electrode, carbonised at 800 °C. The rate capability test of such an electrode shows an invariant response with different current density (Figure 7f). The Coulombic efficiency over 90% in this electrode shows excellent reversibility over variant current rates.

The charge transport across CX electrodes is analysed using EIS experiments (Figure 7g). A low value of the equivalent series resistance (ESR), consisting of the resistances offered by the electrode–electrolyte interface, the electrode and the current collector interface, and the bulk electrolyte resistance, represents a better charge storage [2,6]. This value is found to be lower in CX electrodes, carbonised at 800 °C, indicating the role of pore accessibility from meso to micropores for the transportation of electrolyte ions. A very low electrical conductivity of CX at 500 °C and 600 °C resulted in an increase in overall impedance and hindered the charge transport process. The impedance data are further fitted to an equivalent ladder model circuit (Figure 7h) using the ZsimpWin 3.21 software to understand the relative charge transport in the CX electrodes. The converged values of the fitting parameters are presented in Table 3. The resistance outside the pore network, R_1_, which includes ESR, accounted for the pore accessibility of the ions. Toward the lowest level of the pore hierarchy, the resistance increases, representing the accessibility of electrolyte ions to reach the internal surface of the electrode. The capacitance contribution is greater at the lowest level of pore hierarchy, as most of the surface area for charge storage held theirs. The relative analysis of charge storage in CX electrodes can be easily understood through this table. However, the major difference in leakage resistance is observed in CX electrodes, which mainly account for the differences in electrical conductivity.

Thus, the electrical conductivity is the major influencing factor that differentiates the performance of CXs, obtained at lower carbonisation temperatures of 500 °C and 600 °C from higher carbonisation temperatures of 700 °C to 900 °C. Furthermore, the surface oxygen content clearly distinguished the electrochemical performances from 700 °C to 900 °C, and can thus be considered as a significant factor to decide the performances at higher carbonisation temperatures. Finally, the meso-micropore connectivity as analysed from charge transport kinetics distinguishes the performances of CX electrodes, obtained at 700 °C and 800 °C.

### 3.2. Effect of Type of Activation

For activation, the different extents of CO_2_ exposure and different concentrations of KOH are considered on the CX sample, carbonised at 800 °C due to its better electrochemical performance. The results of the experiment carried out with nitrogen adsorption–desorption show a similar nature of Type IVa isotherms [31] (Figure 8a,b), as obtained before activation. However, capillary condensation begins early at a relative pressure (P/P_0_) ~0.5, indicating the creation of mesopores of smaller width after activation in both types. Furthermore, a higher uptake at low relative pressure compared to that of the CX sample indicates the development of more micropores. The presence of more enhanced micropore characteristics and improved smaller-width mesopores is seen in the DFT pore size distributions (Figure 8c,d). KOH activation is found to be better in creating more micropores smaller than 1.2 nm, compared to CO_2_ activation. In addition, mesopores of sizes ranging from 2 to 4 nm are created after KOH activation. On the other hand, a longer exposure to CO_2_, of 6 h, led to little improvement in the pores in this size range. The mesopore walls are etched during CO_2_ exposure, resulting in narrower pore sizes smaller than 10 nm, compared to the CX sample, carbonised at 800 °C. On the contrary, KOH activation creates wider pore sizes that extended to 24 nm. Thus, KOH activation involves more surface etching than CO_2_ activation. The improvements in both micropore and mesopore characteristics in KOH activation are clearly seen in Table 4. The pore volume or the surface area contributing to micro-mesopores almost doubled after both types of activation.

During activation, the CX network gets etched to generate nanofeatures in activated form, which is clearly understood through SEM micrographs (Figure 9). The carbon grains were reduced to almost half their size. In particular, KOH activation led to a greater decrease in carbon grain size compared to CO_2_ activation. During CO_2_ exposure, the burn-off caused by oxidation leads to pore opening in the CX structure, which further leads to loss of carbon and limits the oxidative environment. The percentage of burn-off was found to be around 65% for CO_2_ exposure up to 6 h under the conditions specified above, in the experimental section. On the other hand, KOH activation involves the intercalation of K^+^ ions and the oxidation of carbon to carbonate ions, that leads to the generation of pores [7,16]. Again, the high activation temperature (750 °C, in this case) responsible for the evolution of CO_2_ from the decomposition of K_2_CO_3_ adds a further contribution to the development of pores in the CX network. Thus, KOH activation provides a combined effect of activation according to the following reactions [7,37], and is superior to the development of nanofeatures in CX.
6 KOH + 2 C → 2 K + 3 H_2_ + 2 K_2_CO_3_(12)
K_2_CO_3_ → K_2_O + CO_2_(13)
CO_2_ + C → 2 CO(14)
K_2_CO_3_ + 2 C → 2 K + 3 CO(15)
C + K_2_O → 2 K + CO(16)

The effect of the activation treatment of the carbonaceous sample (CX-800) by KOH and CO_2_ on the surface chemistry and structure was studied by FTIR spectroscopy (Figure 10) and elemental analysis (Table 4). These two types of carbon activation had a different impact on the resulted surface chemistry and chemical composition of treated carbons, which is clearly noticeable in Figure 10. Such a different behaviour of activation treatment (Figure 10) could be explained by the fact that one treatment method has a chemical impact (KOH) and another a physical impact (CO_2_). Chemical treatment (KOH activation at 750 °C) is related to the affection of the surface chemistry as well as the texture properties of the treated carbon, while physical activation (CO_2_ activation at 800 °C) is assumed to predominantly influence the texture properties of the treated sample. In Figure 10, it is noticeable that the KOH activation results in the creation of some oxygen functional groups. In particular, the band appeared at 1100 cm^−1^, corresponding to the vibration of the –C-O containing groups. Since the temperature used during chemical activation was 750 °C, these possible oxygen groups could be assigned to ethers or traces of phenols. On the other hand, CO_2_ activation has no significant impact on surface chemistry, rather than influencing the porous properties (increase in surface area and pore volume) of treated carbons, which was similarly reported by Elsayed et al. [33].

The electrochemical performances of activated CX electrodes are presented in Figure 11. CV plots (Figure 11a) show a quasi-rectangular shape with a small distortion at peak voltage due to the introduction of oxygen functionality into activated forms. Similar observations of shapes deviation in regular triangular are noted in the CP plot (Figure 11b), confirming the additional contribution of pseudocapacitance to double-layer charge storage in activated CX electrodes. The increased area under rectangular shape in the CV plot or the longer charge-discharge times are observed for activated electrodes, representing higher capacitance values compared to that of CX electrode before activation. The enhanced capacitance results from the suitable micro-mesopore connectivity with an increase in the surface area or the pore volume of corresponding micro-mesopores to almost double the value responsible for double-layer charge storage. In addition, the oxygen functionality favours wettability in reference to the KOH electrolyte and provides some sort of pseudocapacitance as mentioned above in reactions (10) and (11). The KOH-activated CX electrode with a 6:1 ratio shows the longest charge–discharge time, indicating the highest capacitance among all electrodes. The capacitance value is found to be 222 F g^−1^ at a current density of 0.1 A g^−1^. The highest capacitance in this electrode is possible due to higher proportions of micropore features, oxygen content, and wider mesopore sizes (up to 24 nm), providing easier transport compared to all other electrodes, as observed from Table 4 and Figure 8d. However, the micropore feature is the main factor in deciding the electrochemical performance when observing the performance of the KOH-CX-4:1 and CO_2_-CX-6h electrodes in CV and CP plots. These electrodes are further tested with the rate capability test (Figure 11c) showing almost invariant performance with increased current density in KOH-activated CX electrodes compared to CO_2_ activated CX electrodes, with a significant decrease in performance observed for CO_2_-CX-4h electrodes. The absence of pores in the mesopore sizes from 2 to 4 nm in this electrode may be the reason for the inaccessibility of the pores at high current density. The specific capacitance obtained in this study is much higher compared to the values reported in literature for RF derived carbon-gels (aerogels/cryogels/xerogels) synthesised with a solvent exchange step prior to drying for retaining pores [17,38,39,40,41,42,43,44,45], and comparable to the recent studies [46,47,48,49] using KOH electrolyte.

The role of mesopores and the connectivity of pores is further understood through charge transport in such electrodes by impedance analysis (Figure 11d). Higher ESR values are observed for CO_2_-activated CX electrodes compared to KOH-activated CX electrodes. This highlights the role of a wide mesopore size distribution up to 24 nm and a higher volume of pores from 2 to 4 nm created in KOH-activated CX electrodes responsible for a better transport of electrolyte ions towards the newly created micropores smaller than 1 nm for double-layer charge storage. The Ragone plot for activated CX electrodes is shown in Figure 11e. The good rate capability resulted in a much smaller change in the specific energy with an increase in the current density. The accessibility of pores for the electrolyte ions because of the proper pore connectivity in activated CX electrodes is responsible for rapid discharging even at high current density. This leads to an increase in specific power with current density without compromising the specific energy significantly. The specific energy to the constant level of 10 Wh kg^−1^ at a specific power of 0.4 kW kg^−1^ is reached with the KOH-CX-6:1 electrodes. Furthermore, a set of fresh electrodes from the KOH-CX-6:1 sample in a two-electrode cell underwent a cyclic stability test at 1 A g^−1^ for 1000 consecutive charge–discharge cycles shows a capacitance retention of 97% for a corresponding coulombic efficiency of 94%, indicating the excellent stability of the electrode (Figure 11f). Such a performance is undoubtedly favourable for the future development of EC electrodes with the use of a higher charging voltage for EC cells using ionic liquids or organic electrolytes, and the use of such electrodes in an asymmetric or hybrid set-up.

## 4. Conclusions

This article reviews the effect of different carbonisation temperatures and type of activations on the physicochemical properties of CXs and their relation to corresponding electrochemical characteristics for use in EC electrodes. The increase in carbonisation temperature from 500 °C to 900 °C leads mainly to the development of micropore features in CXs. On the other hand, there is little influence of carbonisation temperature on mesopore sizes and volume, and mesopores remained in the size range from 3.5 to 14 nm even with high carbonisation temperatures. The percentage of surface and total oxygen decreases in CXs with an increase in carbonisation temperature, and mainly phenol (-OH) and carbonyl (C=O) surface groups are present in CXs. The hydrophilicity depends not only on the oxygen content but also on the porous texture of CXs. Furthermore, the increase in carbonisation temperature led to an increase in the crystallite diameter, a reduction in the interlayer spacing, and thus the well-organised stacked graphitic layers in CXs, which improved crystallinity to some extent. A very low electrical conductivity is observed at carbonisation temperatures of 500 °C and 600 °C, which increased significantly for higher temperatures, of 700 °C to 900 °C. The electrochemical performances of CX electrodes depend primarily on electrical conductivity, followed by surface oxygen content, and pore hierarchy from meso to micropores. These results suggest that the carbonisation temperature of 800 °C can be chosen as the optimum temperature for making EC electrodes from a carbon precursor containing oxygen groups.

Further, both types of activation protocols using CO_2_ and KOH induce micro-mesopores in CX, which led to an almost two-fold increase in surface area. However, KOH activation is found to be superior compared to CO_2_ activation. KOH activation created more micropores smaller than 1.2 nm, mesopores of size range from 2 to 4 nm, and wider pores of up to 24 nm in CX compared to CO_2_ activation. In addition, a greater decrease in the size of carbon grains, as well as a higher degree of oxygen content, is observed from KOH activation. From both types of activation, it can be concluded that the micropore features are solely responsible for increasing the capacitance values, while the role of mesopores is equally important for better ion and charge transport. The KOH activated CX electrode with a 6: 1 ratio shows a superior capacitance with a good rate capability and is thus favourable for further use in an electrochemical cell with higher charging voltage and in an asymmetric or hybrid mode.

## Figures and Tables

**Figure 1 materials-15-02431-f001:**
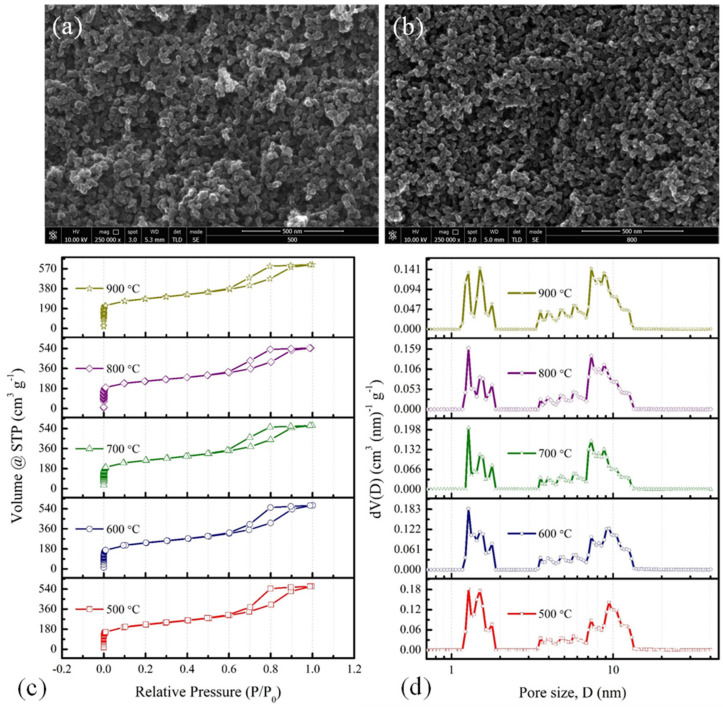
SEM images of CXs at (**a**) 500 °C, and (**b**) 800 °C, (**c**) nitrogen adsorption–desorption isotherms, and (**d**) pore size distributions for CXs obtained at different carbonisation temperatures.

**Figure 2 materials-15-02431-f002:**
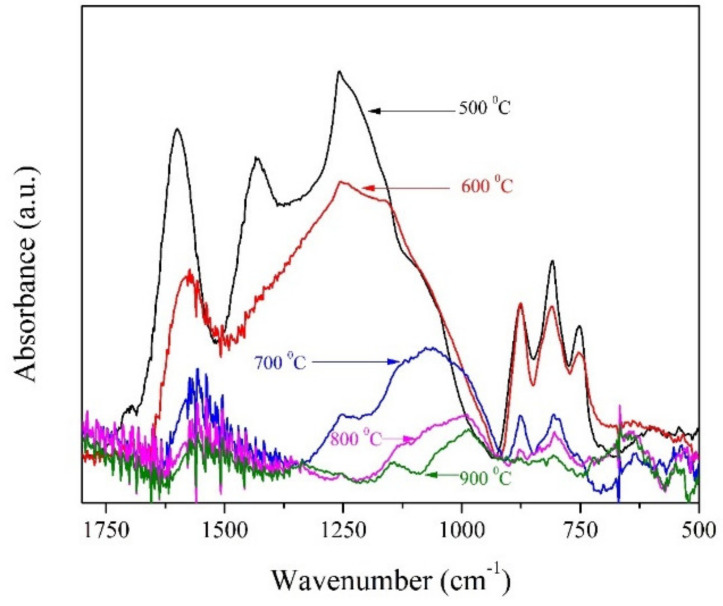
FTIR spectra of CXs obtained at different carbonisation temperatures.

**Figure 3 materials-15-02431-f003:**
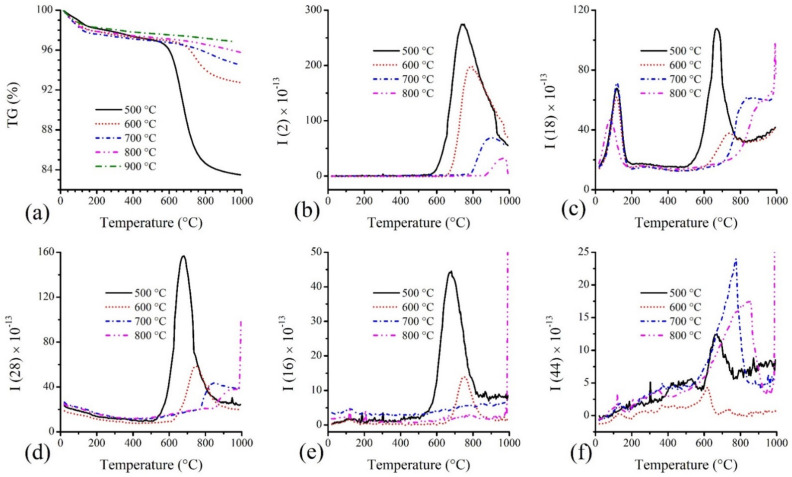
TG-MS curves of CXs obtained at different carbonisation temperatures: (**a**) TG curves and MS signals for (**b**) H_2_, (**c**) H_2_O, (**d**) CO, (**e**) CH_4_, and (**f**) CO_2_.

**Figure 4 materials-15-02431-f004:**
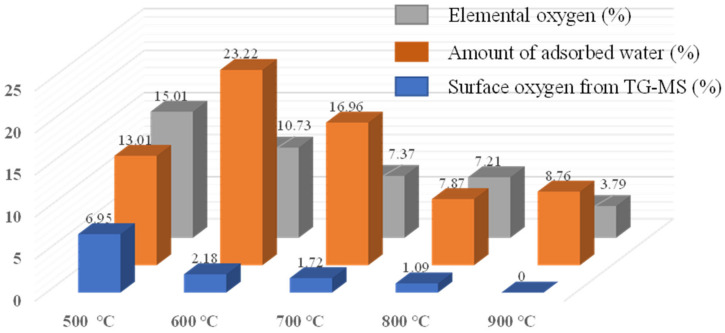
Comparison of surface oxygen from TG-MS analysis with total oxygen and water vapour adsorption.

**Figure 5 materials-15-02431-f005:**
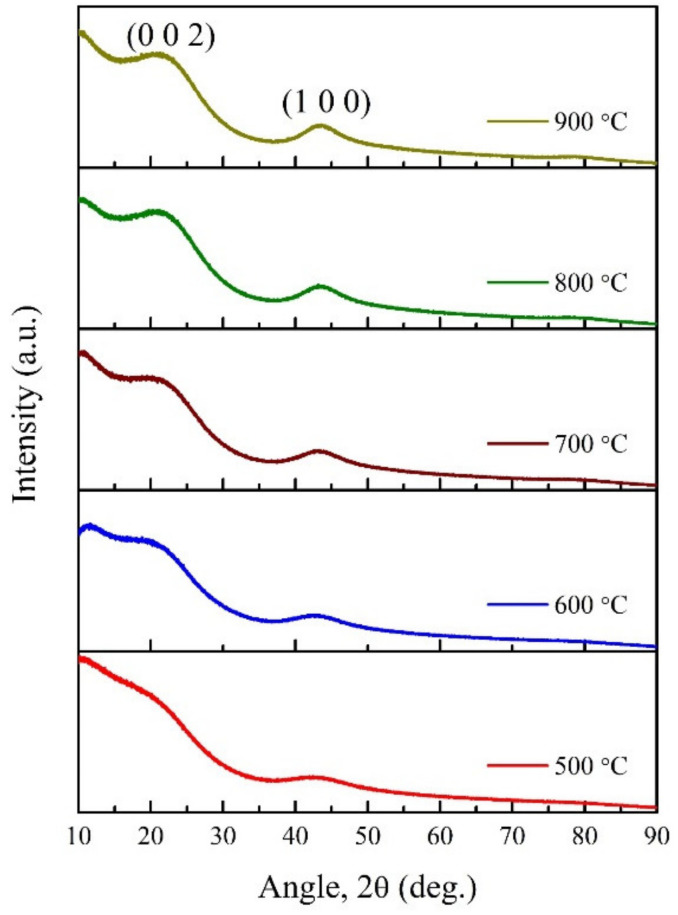
XRD spectra of CXs obtained at different carbonisation temperatures.

**Figure 6 materials-15-02431-f006:**
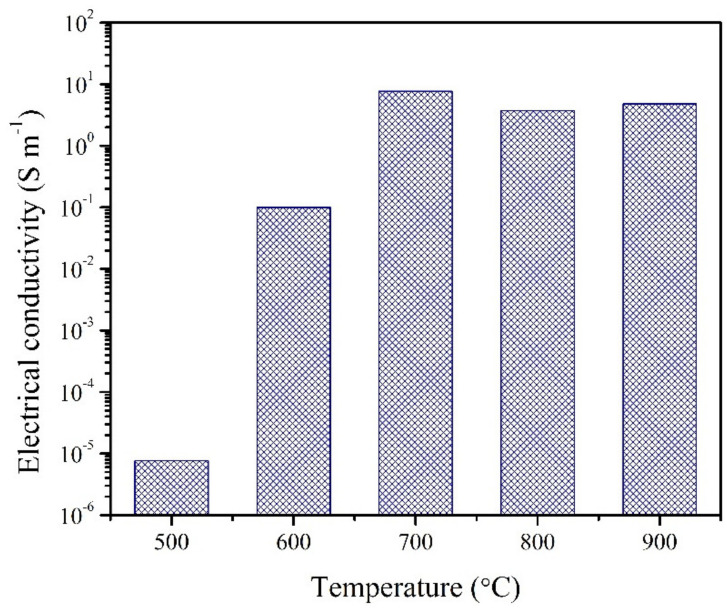
Electrical conductivity of CXs obtained at different carbonisation temperatures.

**Figure 7 materials-15-02431-f007:**
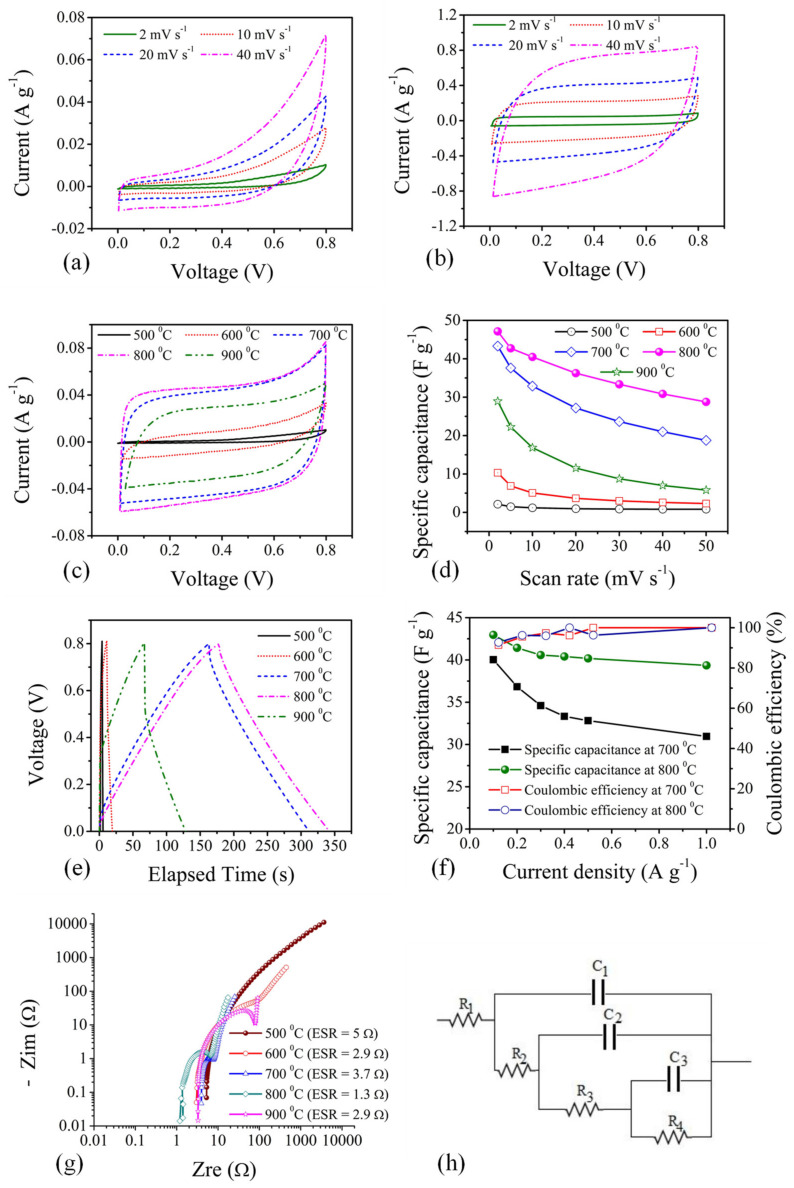
Electrochemical performance of CX electrodes at different temperatures: CV for CX-electrodes carbonised at (**a**) 500 °C, and (**b**) 800 °C, (**c**) CV comparison at 2 mV s^−1^, (**d**) rate capability with scan rate, (**e**) CP comparison at 0.1 A g^−1^, (**f**) rate capability and coulombic efficiency with current density, (**g**) Nyquist plot, and (**h**) equivalent circuit for fitting impedance data.

**Figure 8 materials-15-02431-f008:**
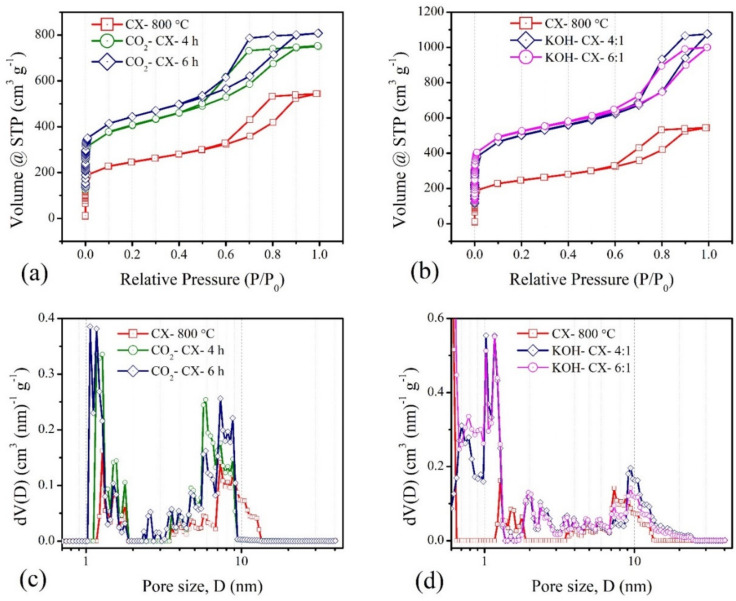
Nitrogen adsorption–desorption isotherms for (**a**) CO_2_ activated, and (**b**) KOH activated CXs, and pore size distributions for (**c**) CO_2_ activated, and (**d**) KOH activated CXs.

**Figure 9 materials-15-02431-f009:**
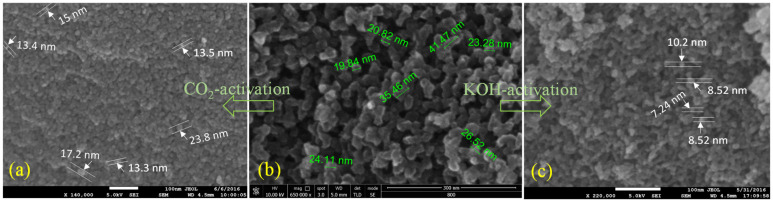
SEM images of (**a**) CO_2_-CX-6h, (**b**) CX carbonised at 800 °C, and (**c**) KOH-CX-6:1.

**Figure 10 materials-15-02431-f010:**
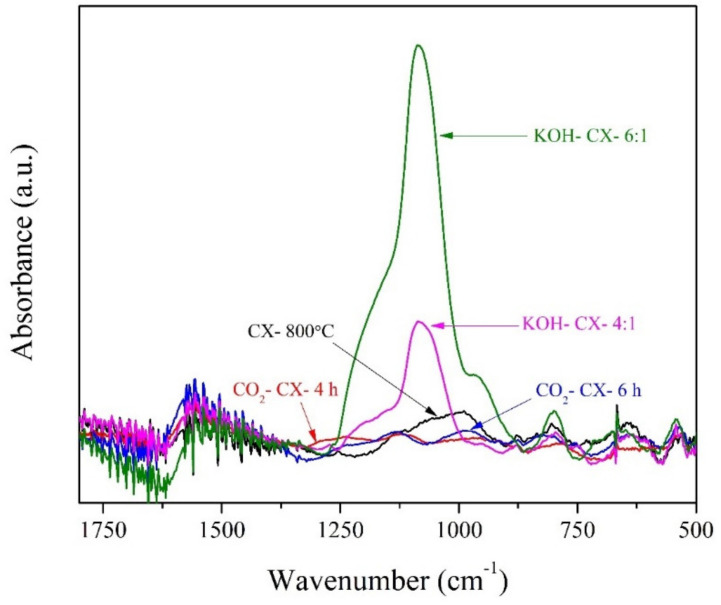
FTIR spectra of CX and its activated forms.

**Figure 11 materials-15-02431-f011:**
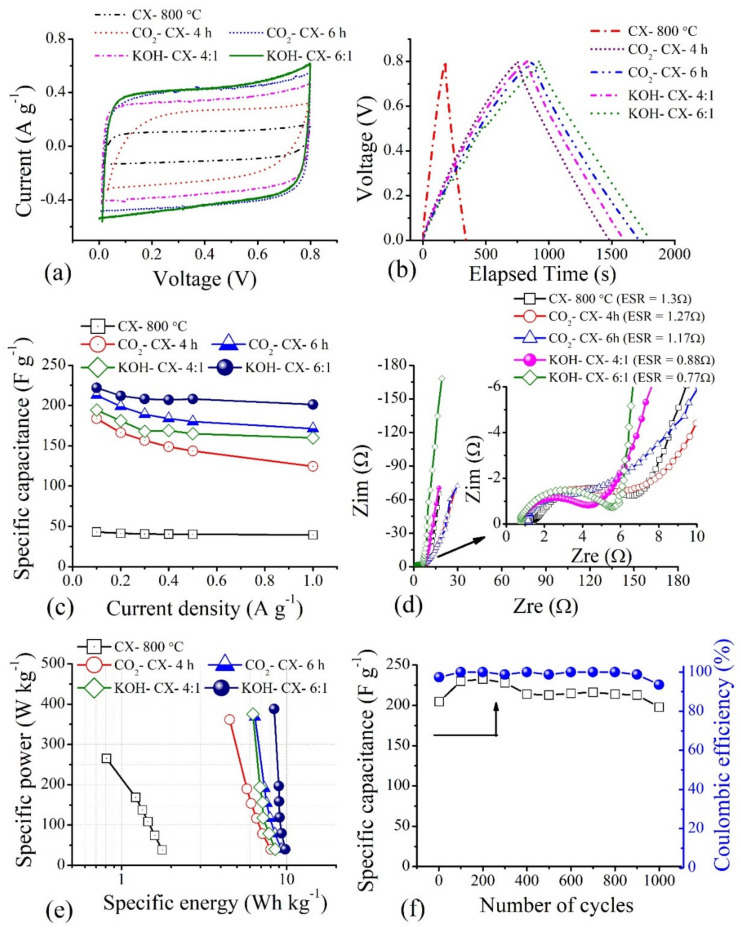
Electrochemical performance of activated CX electrodes: (**a**) CV at 5 mV s^−1^, (**b**) CP at 0.1 A g^−1^, (**c**) rate capability with current density, (**d**) Nyquist plot, (**e**) Ragone plot, and (**f**) cyclic stability at 1 A g^−1^.

**Table 1 materials-15-02431-t001:** Effect of carbonisation temperature on porous texture and elemental compositions of CXs.

Temperature	Micro + Mesopore		Micropore		Mesopore		Elemental Composition from CHNS Analyses		
	V_DFT_	S_DFT_	V_mic_	S_mic_	V_meso_	S_meso_	C%	H%	O%
(°C)	(cm^3^ g^−1^)	(m^2^ g^−1^)	(cm^3^ g^−1^)	(m^2^ g^−1^)	(cm^3^ g^−1^)	(m^2^ g^−1^)			
500	0.86	1016	0.23	722	0.63	294	81.54	3.02	15.01
600	0.87	1116	0.25	821	0.62	295	86.75	2.26	10.73
700	0.87	1276	0.28	975	0.59	301	90.65	1.72	7.37
800	0.83	1219	0.27	944	0.56	275	90.99	1.44	7.21
900	0.93	1389	0.32	1078	0.61	311	94.82	0.68	3.79

**Table 2 materials-15-02431-t002:** Effect of carbonisation temperature on structural and lattice parameters of CXs, calculated from XRD data.

Temperature (°C)	$2\theta_{002}$	$2\theta_{100}$	$d_{002}\;(Å)$	$L_{c}\;(Å)$	$L_{a}\;(Å)$	$N_{av}$	R*
500	21.64	43.09	4.10	10.05	22.38	3.45	1.13
600	21.79	43.14	4.07	10.25	21.12	3.55	1.25
700	22.47	43.45	3.95	9.38	26.09	3.37	1.36
800	22.72	43.72	3.91	9.56	28.36	3.44	1.47
900	22.61	43.44	3.93	9.80	29.83	3.49	1.48

**Table 3 materials-15-02431-t003:** Effect of carbonisation temperature on converged values of circuit-fitting parameters obtained from EIS data for CX electrodes.

CX Electrode-Temperature (°C)	R_1_ (Ω)	C_1_ (F)	R_2_ (Ω)	C_2_ (F)	R_3_ (Ω)	C_3_ (F)	R_4_ (Ω)	Chi^2^
500	5.9	1.3 × 10^−4^	15	3.9 × 10^−4^	495.3	4.9 × 10^−4^	27,890	1.3 × 10^−2^
600	3.7	1.4 × 10^−4^	16.3	5.3 × 10^−4^	108.7	0.014	722	2.9 × 10^−2^
700	4.3	1.9 × 10^−4^	3.6	0.001	14.68	0.145	378	1.1 × 10^−2^
800	1.5	1.9 × 10^−4^	2.9	0.004	3.7	0.197	219	1.9 × 10^−2^
900	3.6	1.4 × 10^−4^	12	5 × 10^−4^	60.6	0.113	174	1 × 10^−2^

R_1_: circuit equivalent series resistance; C_1_: capacitance in first level pore hierarchy; C_2_ and C_3_: capacitances for the intermediate and the lowest hierarchy pores, respectively; R_2_ and R_3_: resistances to charge propagation at the intermediate and the lowest hierarchal pores, respectively; R_4_: leakage resistance associated with C_3_; Chi^2^: goodness of fit.

**Table 4 materials-15-02431-t004:** Effect of type of activation on porous texture and elemental compositions of CXs.

Activation	Micro + Mesopore		Micropore		Mesopore		Elemental Composition from CHNS Analyses		
	V_DFT_	S_DFT_	V_mic_	S_mic_	V_meso_	S_meso_	C%	H%	O%
	(cm^3^ g^−1^)	(m^2^ g^−1^)	(cm^3^ g^−1^)	(m^2^ g^−1^)	(cm^3^ g^−1^)	(m^2^ g^−1^)			
CO_2_-CX-4h	1.17	2157	0.49	1740	0.68	417	82.85	1.77	14.9
CO_2_-CX-6h	1.25	2334	0.52	1889	0.73	445	82.6	1.83	14.93
KOH-CX-4:1	1.66	2354	0.56	1830	1.1	524	77.49	1.93	20.21
KOH-CX-6:1	1.52	2381	0.61	1916	0.91	465	76.46	2.21	20.9

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
