# Peer review of "Revisiting the Effect of Pyrolysis Temperature and Type of Activation on the Performance of Carbon Electrodes in an Electrochemical Capacitor"

_materials, 2022, doi:10.3390/ma15072431_

Round 1
Reviewer 1 Report
Despite the work is not a novel research point as the effect of pyrolysis and activation process on the electrochemical activity of carbon are widely studied before. I have the following comments before its acceptance
- In the abstract section, the value of electrical conductivity (7 × 10-6 – 8 S m-1) is misleading. Do you mean (7 × 10-6 – 8× 10-6) or as it written from 7 × 10-6 to eight.
- In the previous research work, the optimum activation ratio between KOH and Carbon is 2:1. Why did you decide to use the ratios of 4:1 and 6:1?
- In the XRD figure, with increasing the pyrolysis temperature more intense peaks are observed. These intense peaks not only related to the crystallinity but also related to the particle size. These data suggest larger particle size and lower surface area for the carbon xerogel carbonized at 900 °C, and these results in contrary with the results obtained from surface area analysis where carbon at 900 °C has the highest surface area.
- Please mention the model of the used potentiostat.
- The capacitance at 800 °C is higher than that at 900 °C, while the carbon at 900 °C has higher surface area, mesopore and micropore volumes which enhances the ionic diffusion and reduces the electrode resistance. So it is expected that carbon at 900 °C has the highest capacitance. Is the reason just the content of surface oxygen?
- Why you did not calculate the specific capacitances (F/g) from CV and CP curves?.
- In figure 11-d, why the Y-axis of Nyquist plot is in negative scale.
Reviewer 2 Report
The manuscript reported the effect of carbonization temperature and the type of activation from physical to chemical approaches on the properties to understand the resulting performance in ECs. The way of activation using a varied extent of CO2 exposure and KOH concentrations played differently in CX in terms of pore connectivity from meso to micropores and their contributions, degree of oxidation, and resulted in different electrochemical behaviors.
I consider the content of this manuscript will definitely meet the reading interests of the readers of the Materials journal. However, the discussion and explanation should be further improved. Therefore, I suggest giving a minor revision and the authors need to clarify some issues. This could be a comprehensive and meaningful work after revision.
- The length of the abstract is too long, far exceeding the requirements of the journal. ‘Abstract: The abstract should be a total of about 200 words maximum’[https://www.mdpi.com/journal/materials/instructions]. Hence, the content of the abstract needs to be appropriately reduced and refined.
- For the Keywords, I suggest adding ‘electrochemical performance’ and ‘meso-micropore connectivity’ to attract a broader readership.
- For the Introduction part, in the beginning, I suggest that the author first introduce the types of electrode materials in capacitors. After that, the author should emphasize the advantages of carbon-based electrode materials, and then focus on carbon electrodes. At the beginning of the current version directly entering the carbon electrode of the supercapacitor is slightly abrupt.
- Line 72, ‘In addition, a carbonaceous product emerged with different oxygen functionalities that significantly alter acidic-basic characteristics and surface interactions, which are responsible for the adsorption or improved wettability of electrolyte ions [1,2,5].’Here should refer to the increase of hydrophilicity due to oxygen functionalities, as well as the oxygen functional groups providing shorter electrocatalytic pathways [International Journal of Energy Research 44.5 (2020): 3839-3853].
- Line 258, ‘The decrease in percentages of oxygen and hydrogen appeared due to the removal of labile groups during the pyrolysis process. The elemental compositions of these samples seem to be consistent with an increase in the carbonization temperature.’The former sentence describes the decrease of oxygen and hydrogen, while the latter sentence describes that the compositions are consistent. The 'consistent' in the latter sentence seems to easily cause confusion. It is suggested to explain this sentence better.
- Line 327, ‘However, the decrease in values of interlayer spacing with carbonization temperature indicates an increase in the number of stacked graphitic layers organized in the carbonaceous structure.’The graphitization temperature generally exceeds 2000 ℃, so how are the graphitic layers formed? There should be some more details about the mechanism.
- Why only samples treated at 500 and 800 degrees are tested for electrochemical performances in Figure 7? For conductivity value, it is clear that 700 degrees are the highest in terms of conductivity. It is not very clear about the selection of samples for Figure 7.
